SA-FLIDS: secure and authenticated federated learning-based intelligent network intrusion detection system for smart healthcare

Bensaid Radjaa 1
Labraoui Nabila 2
http://orcid.org/0000-0001-5660-0660 Abba Ari Ado Adamou 3
http://orcid.org/0000-0003-3859-859X Saidi Hafida 1
Mboussam Emati Joel Herve 4
http://orcid.org/0000-0001-5360-9782 Maglaras Leandros 5 l.maglaras@napier.ac.uk
1 STIC Lab, Abou Bekr Belkaid Tlemcen University , Tlemcen , Algeria
2 LRI Lab, Abou Bekr Belkaid Tlemcen University , Tlemcen , Algeria
3 LaRI Lab, University of Maroua , Cameroon , Cameroon
4 Department of Mathematics and Computer Sciences, University of Dschang , Cameroon
5 School of Computer Science, Edinburgh Napier University , Edinburgh , United Kingdom
Akleylek Sedat
Electronic publication date: 2024 Dec 13
Publication date: 2024
Volume: 10
Electronic Location ID: e2414
Received 2024 May 28; Accepted 2024 Sep 23
Copyright: © 2024 Bensaid et al.
Copyright year: 2024
Copyright holder: Bensaid et al.
License: This is an open access article distributed under the terms of the Creative Commons Attribution License, which permits unrestricted use, distribution, reproduction and adaptation in any medium and for any purpose provided that it is properly attributed. For attribution, the original author(s), title, publication source (PeerJ Computer Science) and either DOI or URL of the article must be cited.
License URL: https://creativecommons.org/licenses/by/4.0/

Keywords: Intrusion detection, Cybersecurity, Smart healthcare

Funding: The authors received no funding for this work.

==============================
Smart healthcare systems are gaining increased practicality and utility, driven by continuous advancements in artificial intelligence technologies, cloud and fog computing, and the Internet of Things (IoT). However, despite these transformative developments, challenges persist within IoT devices, encompassing computational constraints, storage limitations, and attack vulnerability. These attacks target sensitive health information, compromise data integrity, and pose obstacles to the overall resilience of the healthcare sector. To address these vulnerabilities, Network-based Intrusion Detection Systems (NIDSs) are crucial in fortifying smart healthcare networks and ensuring secure use of IoMT-based applications by mitigating security risks. Thus, this article proposes a novel Secure and Authenticated Federated Learning-based NIDS framework using Blockchain (SA-FLIDS) for fog-IoMT-enabled smart healthcare systems. Our research aims to improve data privacy and reduce communication costs. Furthermore, we also address weaknesses in decentralized learning systems, like Sybil and Model Poisoning attacks. We leverage the blockchain-based Self-Sovereign Identity (SSI) model to handle client authentication and secure communication. Additionally, we use the Trimmed Mean method to aggregate data. This helps reduce the effect of unusual or malicious inputs when creating the overall model. Our approach is evaluated on real IoT traffic datasets such as CICIoT2023 and EdgeIIoTset. It demonstrates exceptional robustness against adversarial attacks. These findings underscore the potential of our technique to improve the security of IoMT-based healthcare applications.

Introduction

The rapid rise of technologies, particularly in artificial intelligence, has significantly influenced various industries including healthcare (Lee & Yoon, 2021). Smart healthcare introduces innovative ideas and architectures that leverage the recent advancements in Information and Communication Technology (ICT), including the Internet of Things, Cloud and Fog computing, wearable devices, Electronic Health Records (EHRs), and more (Kumari et al., 2018). Thus, smart healthcare has the potential to enhance citizen health, deliver exceptional services, reduce healthcare costs, and empower healthcare practitioners to make more precise diagnoses and treatment decisions (Mondejar et al., 2021).

Motivation

Despite the transformative potential of smart healthcare systems, integrating various technologies and medical devices also introduces significant vulnerabilities. Cyber-attacks on IoT medical devices can compromise patient safety, data integrity, and the availability of healthcare services (Djenne & Saïdouni, 2018). These attacks are becoming increasingly sophisticated, and given the limited resources and insufficient computing capabilities of many IoT devices, there is a substantial risk of these devices being exploited as bots for launching further attacks (Bensaid et al., 2024). Therefore, deploying a Network-based Intrusion Detection System (NIDS) is critical to ensure the sustainability and security of smart healthcare-based IoMT applications (Saidi, Labraoui & Ari, 2022), secure patient data, and mitigate the evolving threats posed by cyberattacks. However, these systems tend to generate high false positive rates and lack the scalability and efficiency required to combat emerging cyber threats. The adoption of machine learning approaches within IDS highlights the need for intelligent, anomaly-based detection systems that can operate with minimal human intervention (Radjaa, Nabila & Salameh, 2023). However, these models have common drawbacks, including reliance on a single entity to manage data from all network users, large-scale medical data storage in cloud servers leading to potential single-point failures, and concerns related to centralized data governance, which raise privacy issues (Radjaa, Nabila & Salameh, 2023). To address these challenges, federated learning (FL) offers a promising solution by allowing mobile devices to collaboratively train a shared model while keeping data decentralized (Lim et al., 2020). Several studies have employed FL-based IDS (Iwendi et al., 2021; Schneble & Thamilarasu, 2019) to enhance IoT security while preserving privacy. However, FL is susceptible to adversarial attacks such as poisoning and Sybil attacks, which can distort model accuracy and convergence. Thus, there is a crucial necessity for secure FL mechanisms to guard against manipulated data and models. In this context, the integration of blockchain technology with FL can provide an innovative framework to counter these attacks by ensuring immutability, transparency, and security of the data and model updates (Qu et al., 2020; Ali, Karimipour & Tariq, 2021).

Contributions

To the best of our knowledge, none of the prior research has specifically emphasized user authentication within the FL process. In our article, we propose the SA-FLIDS framework, a novel Secure and Authenticated FL-based NIDS framework using Blockchain for protecting IoMT-enabled smart healthcare networks. SA-FLIDS employs a secure FL approach for detecting anomalies in the IoMT network, thereby creating an intelligent NIDS. This system can effectively identify and counter cyber-attacks aimed at IoMT devices, ensuring the security and integrity of the overall system. It is based on blockchain and Self-Sovereign Identity (SSI) technologies for secure FL, employing an identity management and devices authentication scheme to protect FL against adversarial attacks and ensure that only trusted nodes (Benfriha et al., 2023) contribute to the training. Hence, participants’ privacy is ensured by SA-FLIDS, as the centralized training module does not share users’ private data. SA-FLIDS uses gRPC (Remote Procedure Call) for efficient node communication and TLS (Transport Layer Security) for encrypted channels. It also employs the trimmed mean method for model aggregation, reducing the impact of adversarial data and outliers, enhancing resilience against data poisoning, and maintaining the global model’s integrity. Our proposed model is assessed using the two latest datasets, CICIoT2023 (Neto et al., 2023) and Edge-IIoTset (Ferrag et al., 2022), and demonstrates superior performance across key metrics including accuracy, precision, recall, and F1-score, while maintaining a low rate of false positives. Moreover, we assess and evaluate the blockchain-based client authentication framework, employing the decentralized identifiers (DID) and verifiable credentials (VC) model through the Hyperledger Indy blockchain and Aries library. The overall contributions of this article are outlined below: We propose our Secure and Authenticated Federated Learning-based NIDS (SA-FLIDS) framework to identify and prevent cyber-attacks in IoMT-enabled smart healthcare systems.

We incorporate blockchain-based Self-Sovereign Identity (SSI) to authenticate participants, ensuring that only trusted nodes participate in the Federated Learning (FL) process.

We employ the trimmed mean method aggregation in FL. This approach enhances resilience against data poisoning attacks and preserves the integrity of the global model in a distributed IoMT environment.

We implement gRPC with TLS encryption in our work specifically for secure communication in an FL-based NIDS for healthcare IoT. This combination ensures both efficient and secure data exchange between IoMT devices, fog nodes, and servers.

We provide a comprehensive evaluation of our proposed system using two recent, real-world IoT security datasets (CICIoT2023 and Edge-IIoTset), demonstrating its effectiveness in detecting a wide range of attacks relevant to smart healthcare environments.

Related work

This section provides an overview of relevant literature on federated learning (FL), machine learning (ML), and blockchain in the context of Intrusion Detection Systems (IDS) for IoT networks.

Schneble & Thamilarasu (2019) proposed FLIDS, an FL-based IDS for medical cyber-physical systems (MCPS). Their model reduces communication and computation expenses and is effective in identifying various attacks. However, it is vulnerable to poisoning attacks.

Similarly, Chatterjee & Hanawal (2021) applied FL with a convolutional neural network for IoT intrusion detection. Their model handles non-IID data and dynamically optimizes through weighted client aggregation. However, it lacks real-world applicability and security considerations, making it vulnerable to network poisoning attacks.

The FL anomaly detection system presented by Man et al. (2021) uses GRUs with preprocessing and ensemble learning, outperforming traditional approaches. However, their model lacks security measures when sharing trained models, leading to a risk of data leakage.

Rey et al. (2022) proposed an FL-based malware detection method for IoT devices but overlooked channel security, reducing system resilience against attacks. Adversarial machine learning algorithms could undermine their effectiveness in IoT healthcare systems.

Ruzafa-Alcázar et al. (2021) integrated differential privacy techniques into training an IDS for industrial IoT using FL. While offering privacy guarantees, it is susceptible to inference attacks, impacting overall performance. Similarly, Zhao et al. (2019) introduced the MT-DNN-FL (Multi-Task Deep Neural Network in FL), demonstrating high detection rates and reduced training time. However, further optimization is needed to accommodate IoT device limitations. Friha et al. (2022) presented FELIDS, a FL-based IDS for securing IoT infrastructures, which uses local learning and the FedAvg algorithm, a widely adopted aggregation method (McMahan et al., 2017). The global model, as computed in previous studies (Schneble & Thamilarasu, 2019; Wang et al., 2022; Elayan, Aloqaily & Guizani, 2021; Wu et al., 2020) shares model updates between devices and an aggregation server. While offering cost-effective computing, challenges such as communication latency and privacy vulnerabilities require more advanced aggregation techniques.

Ashraf et al. (2022) proposed a blockchain-based FL IDS for IoT healthcare, with sensor monitoring and an artificial neural network (ANN) model for attack detection. Outperforming existing methods. However, privacy concerns and the decentralized nature of patient data need to be addressed.

Preuveneers et al. (2018) integrated FL with a blockchain featuring access control for IDS. The study’s model is relatively simple, limiting generalization. Similarly, Lakhan et al. (2022) introduced the FL-BETS framework, safeguarding privacy and detecting fraudulent activities using FL and blockchain technology. Computation overhead is a challenge, requiring optimized FL algorithms for IoHT devices. Baucas, Spachos & Plataniotis (2023) proposed a fog-based IoT platform using federated learning and blockchain to enhance privacy and security in wearable healthcare devices. Their system maintains patient privacy and provides robust access control, demonstrated through a custom testbed. However, the study’s testing with only a single dataset type does not focus specifically on securing the IoMT network. Furthermore, it does not specifically deal with adversarial attacks on the federated learning process.

Sindhusaranya et al. (2023) proposed a privacy-preserving approach using FL-BEPP (Federated Learning with Blockchain-Enabled Privacy Preservation) to address both soft and hard constraints in fraud prevention and security for the Internet of Medical Things (IoMT). Their method aims to enhance data privacy and security in healthcare systems. However, the implementation of blockchain transactions and federated learning model updates introduces additional computational overhead, potentially resulting in increased latency and reduced performance, especially in large-scale IoMT deployments. This trade-off between enhanced security and system efficiency presents a challenge for widespread adoption in complex healthcare networks.

Begum et al. (2024) proposed BFLIDS, a system combining blockchain and federated learning for intrusion detection in IoMT networks. The approach uses decentralized model training to preserve privacy, blockchain for secure record-keeping, and smart contracts for system management. They modified the FedAvg algorithm to improve accuracy and resilience against attacks. While BFLIDS showed competitive performance, the study didn’t address the resource constraints of IoMT devices or the potential computational overhead from smart contract integration.

The primary goal of the articles mentioned is to establish an IDS-based federated learning framework to maintain security and preserve privacy in IoT applications. However, many existing solutions encounter challenges in addressing specific potential attacks, such as poisoning, Sybil, Data Tampering, and Eavesdropping attacks within FL-based IDS, particularly in healthcare systems. Ensuring security and privacy in efficient FL-based IDS remains a crucial concern, which is the focal point of our investigation in this study.

To address these challenges, we introduce a new Secure and Authenticated FL-based NIDS framework using Blockchain (SA-FLIDS) for fog-IoMT-enabled smart healthcare systems. The SA-FLIDS framework leverages FL to train a shared prediction model while maintaining decentralized data on the devices themselves and incorporates a blockchain-based SSI model for a privacy-preserving authentication scheme. This combination ensures that only trusted IoMT devices contribute to the FL process. Moreover, our framework distinguishes itself from existing literature by incorporating a robust aggregation method, specifically the trimmed mean, to reduce the influence of outliers or malicious participants while computing the global model. Furthermore, we use TLS encryption combined with secure communication protocols like gRPC which helps ensure data integrity during transmission between devices, fog nodes, and servers.

Background

This section provides essential contextual background for our proposed model. Initially, we introduce FL-based NIDS. Subsequently, we delve into Blockchain-based SSI techniques that are pertinent to and integrated within our proposed framework.

Federated learning for IoT intrusion detection

Federated learning offers enhanced privacy and security in IoT networks by minimizing data transmission (Radjaa, Nabila & Salameh, 2023). In IDS, this approach enables the development of more intelligent machine learning models exposed to diverse data sources while ensuring user privacy (Sarhan et al., 2023). In this process, models are downloaded and updated locally on IoT devices using their data, then transmitted to a central server for aggregation, resulting in an improved global model. However, effective data distribution presents practical and technical challenges for successful federated learning deployment.

Blockchain-based SSI

Now shifting our focus to Blockchain-based SSI, this section explores the utilization of Blockchain technology in establishing secure and decentralized identity management systems.

Blockchain

Is a distributed ledger technology for data transmission and storage that records the history of all transactions. Blockchain has been involved due to its decentralization, immutability, and persistence properties in the distributed peer-to-peer (P2P) network. It utilizes asymmetric cryptography to ensure transactions are done safely (Saidi et al., 2022). Blockchain is based on several elements used to create a secure, transparent, and decentralized system. Each element plays a crucial role in maintaining the integrity and functionality of the blockchain (Mboussam Emati & Mboussam, 2023), including block, transaction, consensus mechanism, smart contract, mining, immutable ledger, cryptographic keys, and hash function.

Self-sovereign identity

is a decentralized approach and a new model for digital identity. Self-sovereign identity (SSI) aims to empower individuals to possess and control digital proof of their credentials. Thus, it helps to prove who we are by establishing trusted relationships to access information (Saidi et al., 2022). SSI is based on two main standards: Decentralized identifier (DID) and verifiable credential (VC): Decentralized identifier is a new type of identifier, defined by the W3C (Emati, Mboussam & Tchendji, 2023). DIDs are designed to enable Self-Sovereign Identity on the internet, providing a way for individuals to have control over their own digital identity without the need for a central authority. Thus, users can selectively disclose only the necessary information for a particular transaction or interaction (Saidi et al., 2022).

Verifiable credential is a standard method for digitally expressing credentials in a cryptographically secure way. It can include metadata, claims, and proofs used to verify a credential (Thomas, Ramaguru & Sethumadhavan, 2022). Credentials are created and signed by the issuer using his private key and then issued to the holder, enabling the verifier to confirm the VC. A holder keeps and shares the received credentials with a verifier. The verifier accepts and approves these credentials (Figueroa-Lorenzo, Benito & Arrizabalaga, 2021). based on the public key associated with a DID.

Cyber attacks and risks in iomt-enabled smart healthcare systems

Due to the critical nature of patient well-being, reliable and secure communication is vital in smart healthcare (Lee & Yoon, 2021). Medical IoT devices, with resource constraints like poor battery life and limited memory, are vulnerable to hacking attempts, potentially integrating them into botnets. Common cyberattacks on compromised IoMT devices include: DoS and DDoS attacks aim to undermine IoMT availability, with DoS using a single botnet and DDoS utilizing multiple botnets (Bensaid et al., 2024).

Information gathering attacks collect comprehensive data about IoMT, often using reconnaissance-like scanning attacks (Jensen, Gruschka & Herkenhöner, 2009).

Exploiting web-based vulnerabilities targets web services on IoMT devices, employing methods like injection, hijacking, and DoS (Jensen, Gruschka & Herkenhöner, 2009).

Communication spoofing attacks enable unauthorized access to network traffic, facilitating data theft and malware dissemination (van der Merwe et al., 2018).

Brute-force attacks attempt to discover passwords or passphrases by iteratively trying words from predefined lists (Stiawan et al., 2019).

The Mirai attack, a widespread DDoS assault, specifically targets IoMT devices (Gamblin, 2017).

Therefore, if a malicious IoMT device compromises a fog server, unauthorized access to sensitive patient data and EHRs will be possible, affecting the data privacy and security of patients. This compromised data often includes private and sensitive information such as credit card details, health conditions, and other confidential data, thereby exposing patients to significant risks. Additionally, the unavailability of fog servers disrupts essential healthcare services such as the monitoring of patient’s vital signs. As a consequence, the ability to track and monitor essential health indicators in real-time is compromised, posing potential risks and presenting challenges in delivering timely and suitable care, as illustrated in Fig. 1. Additionally, in critical situations, such as emergencies, the heightened risk to patient’s lives, underscores the severity of potential consequences. Therefore, it is crucial to establish robust cybersecurity measures and implement comprehensive security protocols to protect patient data which are essential for the sustainability of critical healthcare services and the effective mitigation of risks.

Figure 1 Cyber-attacks in smart healthcare.

Icon credit: Electronic Records icon (mcmurryjulie, https://pixabay.com/fr/vectors/pronostic-ic%C3%B4ne-le-dossier-du-patient-2803190/, Pixabay license); Doctor icon (Freepik, https://www.flaticon.com/free-icon/doctor_5065189, Flaticon license); Nurse icon (Freepik, https://www.flaticon.com/free-icon/nurse_9133509, Flaticon license); Patient icon (Freepik, https://www.flaticon.com/free-icon/patient_4228704, Flaticon License).

Sa-flids system architecture and design goal

In this section, we discuss the SA-FILDS architecture and the threat model, followed by the design goals of the SA-FILD system.

SA-FLIDS system architecture

SA-FLIDS system brings a novel Secure and Authenticated Federated Learning-based NIDS framework for Smart Healthcare using Blockchain Technology, secure communication protocols, DID, and VC. SA-FLIDS system aims to examine network traffic, identify and mitigate cyber-attacks against IoMT devices, and enhance the security of smart healthcare systems. The architecture of our proposed model comprises three layers: the Cloud layer, the Fog layer, and the IoMT devices layer, as depicted in Fig. 2.

Figure 2 The proposed model.

Icon credit: Hospital icon (Freepik, https://www.flaticon.com/free-icon/hospital_4320350, Flaticon license); Healthcare device icon (Smashicons, https://www.flaticon.com/free-icon/healthcare-device_2904470, Flaticon license); Blockchain icon (jojooid, https://www.flaticon.com/free-icon/blockchain_8757988, Flaticon license).

Cloud layer: This layer provides an underlying infrastructure and resources that enable the provision of on-demand and adaptable services accessible from any location (Saidi, Labraoui & Ari, 2022). Fog nodes and fog servers can dynamically allocate resources based on their requirements during the FL process. Fog network layer: The intelligent NIDS based on FL and blockchain is deployed in the Fog layer. This layer supervises the network and makes decisions regarding traffic flow classification. It can be deployed at each hospital or clinic. The Fog Network Layer consists of two distinct sub-layers:

Fog server: Serving as the central server that initiates and constructs a shared global model architecture among participating fog nodes. It is responsible for adding verified model weights to the blockchain, ensuring that local model updates are securely and transparently aggregated and shared across the network using the TLS protocol (Möller et al., 2022). Additionally, blockchain is deployed for identity management to ensure that only authenticated devices can participate in the FL.

Fog nodes: These distributed nodes consist of physical components such as mobile devices, gateways, routers, and switches. They act as clients in FL, performing local model training to protect sensitive medical data. Therefore, fog nodes are effective for local model training due to their proximity to the fog server and the IoMT layer, as well as their increased processing power, memory, and connection as compared to individual IoMT devices. Furthermore, each fog node is identified by a DID which is registered into the blockchain. DIDs are used to sign documents or transactions, create secure and persistent communication channels, and send encrypted private messages (Figueroa-Lorenzo, Benito & Arrizabalaga, 2021).

IoMT layer: The Internet of Medical Things (IoMT) layer is used to sense, collect, encrypt, and upload medical data to the fog nodes for private local model training. The transmitted data can encompass both benign network traffic and potential cyber-attack classification. Moreover, each IoMT device is identified by a unique DID registered on the blockchain to handle the authentication process.

Challenges and design goal

FL systems are susceptible to various adversarial attacks, posing risks to their security, and integrity, and hindering their deployment in NIDS. We detail the following attacks that can be launched against FL systems: Sybil attack, a malicious participant creates multiple fake identities to disrupt FL, injecting biased or misleading information into aggregated model updates (Lian et al., 2023).

Eavesdropping attack, involves unauthorized interception of communication between FL participants, potentially leading to privacy breaches by accessing sensitive information such as model updates or raw data (Lian et al., 2023).

Data poisoning attack, in FL involves injecting adversarial data into the training set of participating devices, aiming to compromise the integrity and performance of the global model (Lian et al., 2023).

Data tempering involves unauthorized modification of data in the FL process, aiming to compromise the integrity and reliability of the global model by injecting malicious or false information.

To tackle the challenges mentioned above, we are emphasizing the following design goals: Ensuring the security of FL.

Ensuring that only authenticated IoMT devices can participate in the FL using blockchain-based DID to prevent Sybil attacks.

Secure communication between IoMT devices, fog nodes, and fog servers, and prevent data tampering and eavesdropping during communication between nodes.

Reduce the influence of outliers or malicious participants when computing the global model using a robust aggregation function to mitigate the impact of adversarial participants engaging in poisoning attacks and introducing noisy data.

Sa-flids system

This section presents details of our proposed scheme, SA-FLIDS, which aims to ensure security and privacy preservation in an IoMT-enabled smart healthcare system.

SA-FLIDS identification and authentication approaches

Figure 3 illustrates a sequence diagram of our system throughout its lifecycle, involving nodes, a fog server (FS), and a blockchain (BC). The process contains three phases: Initialization phase: Nodes and the FS request registration from the BC.

The BC responds by sending a DID, which includes a pair of keys and a VC scheme for each entity.

Mutual authentication phase: Nodes send their VC to the FS, and the FS also sends its VC to the nodes. Both parties then request verification from the BC.

The BC verifies the credentials and, if everything is in order, provides an authentication token to both the nodes and the FS.

Federated learning process phase: The FS sends encrypted data, secured with its private key, to the Nodes, enabling them to work on the data securely.

After processing, the nodes send encrypted updates, secured with their private keys, back to the FS. The FS then aggregates these updates.

Figure 3 Sequence diagram of our proposed SA-FLIDS.

The SA-FLIDS model integrates a Blockchain-based DID and VC system to enhance participant identification within the FL process. Figure 4 demonstrates the triangle of trust and privacy in digital interactions, comprising three main parties: the issuer, holder, and verifier. In this model: The issuer is the hospital, which manages the blockchain. The blockchain stores only the DIDs and VC schemes.

Nodes and FS act as both holders and verifiers depending on their role in mutual authentication. When the fog server receives data from nodes, it acts as the verifier.

When nodes receive an authentication query from the FS, they act as the verifier, and the FS is the owner.

Figure 4 Trust triangle for VC.

Icon credit: Blockchain icon: (© Abdul Basit Noohani Dreamstime.com, https://www.dreamstime.com/visualize-power-blockchain-symbolic-icon-representing-secure-digital-ledgers-decentralized-transactions-image291382086); Distributed Ledger icon (Kalashnyk, https://www.flaticon.com/free-icon/documents_12864624, Flaticon license).

Initially, the issuer uploads proofs and stores participants’ DIDs and schema on the blockchain ledger. Then, the holder receives and stores the issued credentials. Subsequently, the holder presents the credentials to the verifier, who reads and verifies them. This system ensures secure, private, and verifiable digital interactions.

Detectin process FL in SA-FLIDS system

After successful authentication and verification processes, the fog server sets up a global model architecture distributed across the involved fog nodes. During each training round, every fog node updates its local model by conducting training on the data collected from IoMT devices within its proximity. Subsequently, these fog nodes transmit their updated models back to the central fog server. Then, the central fog server collects and aggregates these models from all fog nodes using the robust trimmed mean aggregation. Next, the fog server forwards the updated global model back again to all the fog nodes. This iterative process is repeated for each training round until the final global model is obtained and ready for use. Moreover, all those communications occur securely by using both the gRPC framework for effective communication among nodes and the TLS to ensure end-to-end encryption in the communication channels. The final global model is then used for the detection classification process which is deployed in the NIDS to differentiate between normal and potentially malicious traffic patterns. Algorithm 1 demonstrates the FL process.

Algorithm 1 Federated learning.

1: Initialize global model θ0	
2: for t=1 to T do	
3:   for each fog node i do	
4:    Collect local data Di from IoMT devices	
5:    Train local model θti on Di	
6:    Send θti to server	
7:   end for	
8:   Aggregate local models: {θt1,θt2,...,θtN}	
9:    θt+1=TrimmedMean({θt1,θt2,...,θtN})	
10:   for each fog node i do	
11:    Distribute θt+1 to fog node i	
12:   end for	
13: end for	
14: Output: Final global model θT	

Trimmed-mean aggregation method

The Trimmed-Mean method enhances the accuracy and reliability of the global model in FL by mitigating the influence of outliers and malicious participants. It removes a certain percentage of extreme values from local models and calculates a weighted average of the remaining models. The Trimmed-Mean formula is:

(1) Trimmed−Mean=∑i(wi⋅mi)∑iwi

where:

wi represents the weight of the i-th local model.

mi represents the i-th local model in the trimmed set Mt.

Intelligent NIDS-based SA-FL for mitigation process

In the SA-FLIDS model, The fog server plays a dual role in the system, not only detecting potential intrusions but also promptly and effectively responding to mitigate or minimize the consequences of these attacks. Hence, the SA-FL model which is trained through the robust process, is integrated into the NIDS and incorporates the intrusion detection and response mechanism into the fog server’s architecture. The flow involves continually monitoring fresh incoming traffic and sending it through the FL model for prediction, which has previously been trained, to differentiate between normal and potentially malicious traffic patterns. If the FL model predicts normal traffic, access is granted, demonstrating the model’s ability to make real-time decisions based on learned patterns of normal behavior. Otherwise, if the FL model identifies incoming traffic as indicative of an intrusion or attack, it triggers an alarm in the system monitoring, promptly alerting the system to the potential threat. Therefore, the NIDS responds in real-time, taking proactive measures to block and drop malicious packets, as illustrated in Fig. 5.

Figure 5 SA-FLIDS detection and mitigation scheme.

Experiments and results

This study explores the potential of SA-FLIDS in detecting intrusion in IoMT networks. As a result, this decentralized approach could be crucial for securing healthcare applications. In this section, we present the experimental setup along with the evaluation metrics and results.

Experimental setup

In this section, we delve into the experimental setup for both FL and blockchain-based SSI.

Experimental setup for federate learning

Table 1 presents the parameters used in federated deep learning. In our study, we conducted experiments deploying our model with client sets denoted as K, where K = 10. We employed the Independent and Identically Distributed (IID) approach, ensuring that the data distribution across the dataset matches the distribution of data for each client. Furthermore, To prevent overfitting, the following techniques were used: Stratified K-fold cross-validation: Set k = 5 to split the data into five subsets, evaluating the model’s performance.

L2 regularization: Applied with a factor of 0.01 to the dense layers, adding a penalty to the loss function based on the weights’ magnitude, simplifying the model.

Early stopping: With the patience of three, training stops if the validation loss does not improve, monitoring the validation set’s performance.

Table 1 Federated deep learning classifier parameter setting.

	Parameter	Value	
Federated deep learning classifier	Local epoch	10	
Global epoch	4	
Batch size	128	
Hidden layer	2	
Hidden nodes	128,64	
Activation function	Relu	
Regularization	L2	
Classification function	Sigmoid/softmax	
Optimizer, learning rate	Adam, 0.001	
Loss function	Binary_crossentropy Categorical_crossentropy	

Experimental setup for blockchain-based SSI

The simulation is carried out on a Lenovo ThinkPad P51 -Core i7 2.9 GHz–SSD 1 To–32 Go, computer running Ubuntu 18.04 LTS. The simulation environment is based on Hyperledger Indy (Banerjee et al., 2022), a framework dedicated to self-sovereign identity management. It offers an abstraction that enables DIDs and VCs to be created, verified, and revoked. It embeds Hperledger ursa, a module that provides all the primitives for cryptographic operations. We also use Hyperledger Aries (Manoj, Makkithaya & Narendra, 2022). This is a library for creating agents that can manage the cryptographic wallet of each player. It also offers interfaces for creating functionalities that realize the behavior of the wallet owner.

The simulation begins with the initialization phase, during which 10 nodes plus the server are each initialized in a Docker container as shown in Fig. 6. The server is the trusted authority that manages the blockchain. It creates the genesis block and initializes the chain. Each node is registered on the chain and receives a DID and an authentication scheme. Of course, each DID is accompanied by its key pair, which is stored in each node’s wallet, and the public keys are known to everyone. Next comes the authentication phase, when each node sends its VC to the server, which verifies it and receives the server’s VC (mutual authentication). The principle is to request the blockchain’s response to the VC presented to it. Once each node’s authenticity has been verified. The nodes can exchange data. Remember that confirmation of a VC leads to validation of a token, which has a lifetime (that of the session). Each node in possession of its token can then use it for future exchanges. After authentication, the server sends the chunks of the dataset to the various nodes using its private key as demonstrated in Fig. 3. The nodes receive them and can proceed with processing. The different scores are exchanged using the same process. The communication protocol in this phase is gRPC. However, the solution would work with any other protocol, with security being guaranteed by the authentication token.

Figure 6 Blockchain network.

Icon credit: Blockchain icon (© Stockoxinoxi | Dreamstime.com, https://www.dreamstime.com/blockchain-concept-symbol-vector-icon-image222346576).

Datasets description

Datasets play an essential role in both training and assessing IDSs within IoT networks. The choice of suitable datasets tailored to particular tasks holds significant importance, particularly in evaluating the efficacy of FL approaches for IoT networks. In our experiment, we incorporated two recent datasets specifically designed to mimic real-world conditions for IDSs: CICIoT2023 (Neto et al., 2023), made available in 2023, and the Edge-IIoTset dataset, released in 2022. CICIoT2023 dataset: A Neto et al. (2023) novel and extensive IoT attack dataset to foster the development of security analytics applications in real IoT operations. To accomplish this, 33 attacks are executed in an IoT topology composed of 105 devices, and all attacks are executed by malicious IoT devices targeting other IoT devices. We analyzed a dataset containing 47 features (not including label and sublabel) based on 2,366,956 samples extracted from the first 10 CSV files provided by the Canadian Institute.

Edge-IIoTset dataset: It is tailored specifically for IIoT and IoT applications (Ferrag et al., 2022), providing an authentic test environment closely resembling real-world IoT/IIoT settings. Within this environment, we conducted simulations of genuine cyberattacks to collect datasets comprising both legitimate and malicious network traffic. This dataset includes data generated by various IoT devices, spanning from heart rate sensors to flame sensors, temperature, and humidity sensors. The testbed is structured into seven interconnected layers. We utilized the Selected dataset for ML and DL/DNN-EdgeIIoT-dataset CSV file (Banerjee et al., 2022), which contains 61 features and 2,219,201 samples, encompassing both normal traffic and 14 distinct attacks in the IoT and IIoT environment.

Preprocessing

The datasets undergo several preprocessing steps to ensure their suitability for analysis as demonstrated in Fig. 7. After cleaning the data we first, address imbalanced data by implementing SMOTE (Synthetic Minority Over-sampling Technique) and under-sampling techniques to enhance predictive performance, particularly for minority classes. Secondly, data transformation is conducted using the StandardScaler for standardization, adjusting data to have a mean of 0 and a standard deviation of one. Additionally, feature importance analysis is performed using insights from random forest and XGBoost experiments. Finally, the processed dataset is split into an 80% training set and a 20% testing set, ensuring no duplication between the two, contributing to refining the dataset for subsequent analysis and modeling tasks. In the case of the CICIoT dataset, we opt to eliminate the Brute and Web attack labels due to their limited number of samples, which could potentially skew the analysis and compromise the reliability of the results. The detailed features selected and the attacks used are outlined in Table 2 provided below.

Figure 7 Data preprocessing.

Table 2 Datasets description for experimental evaluation.

	CICIoT2023	Edge-IIoTset	
Features selected	‘flow_duration’, ‘Header_Length’, ‘Protocol Type’, ‘Rate’, ’Srate’, ‘syn_count’,‘urg_count’,‘rst_count’, ‘Tot sum’, ‘Min’, ‘Max’, ‘AVG’, ‘Tot size’, ‘IAT’, ‘Magnitue’, ‘Variance’	‘http.content_length’, ‘http.request.method’, ‘http.referer’, ‘http.request.version’, ‘tcp.ack’, ‘tcp.ack_raw’, ‘tcp.checksum’, ‘tcp.flags’, ‘tcp. len’, ‘tcp.seq’, ‘udp.time_delta’, ‘dns.qry.name.len’, ‘mqtt.conack.flags’, ‘mqtt.protoname’, ‘mqtt.topic’	
Label	‘Benign’, ‘DDoS’, ‘DoS’, ‘Mirai’, ‘Recon’, ‘Spoofing’	‘DoS/DDoS’, ‘Information gathering’, ‘Injection’, ‘Malware’, ‘Man in the middle’, ‘Normal’	

Evaluation metrics

In this section, we introduce the metrics employed in our experiments to evaluate both FL and SSI-based DID.

Metrics used for federated learning evaluation

When conducting intrusion detection using federated deep learning performance analysis, the most common metrics used are: True negatives (TN): Benign network activity correctly classified as normal.

True positives (TP): Malicious network activity correctly identified as an attack.

False positives (FP): Benign network activity incorrectly classified as malicious.

False negatives (FN): Malicious network activity is incorrectly classified as normal.

Moreover, we have used a variety of measures to evaluate our proposed model, including precision, recall, precision, F-score, and accuracy, to conduct a systematic comparative analysis with other relevant approaches as demonstrated in Table 3.

Table 3 Performance metrics.

Metric	Formula	Description	
Accuracy	TP+TNTP+TN+FP+FN	A measure that quantifies the proportion of instances correctly classified among the total number of observed samples.	
Precision	TPTP+FP	A metric that indicates the proportion of correctly predicted positive instances out of the total predicted positive instances.	
Recall	TPTP+FN	The proportion of correctly identified positive samples.	
F-measure	2×Precision×RecallPrecision+Recall	The harmonic mean of precision and recall.	

Metrics used for blockchain-based SSI evaluation

To compute the metrics outlined below, we utilize the following formulas: Startup duration (SD): The duration for the system to initiate.

Connect duration (CD): The time required for the system to establish connections between nodes and the Fog server.

Publish duration (PD): The duration for the system to publish schema credentials and related settings.

Issuing credential duration (ICD): The time taken for the system to issue credentials.

Completed credential exchanges duration (CCED): The total time needed for all credential exchanges to conclude.

Metrics such as SD, CD, PD, and ICD are used to assess the Initialization phase, while CCED is used to assess the Mutual Authentication phase.

Average time per credential duration (ATCD): The average time taken to issue a single credential.

(2) ATCD=ICDNcredentials

where:

ICD is the issuing credential duration.

Ncredentials is the total number of credentials issued.

Average time per transaction duration (ATTD): The average time taken per transaction.

(3) ATTD=∑i=1NtransactionsTiNtransactions

where:

Ntransactions is the total number of transactions.

Ti is the duration of each transaction i.

Evaluation results

We utilize federated deep learning-based NIDS models to detect cyber-attacks in IoMT environments, specifically focusing on the networks of healthcare applications. Our training incorporates the most recent datasets for IDS, including CICIoT2023 and the Edge-IIoTset dataset. We conduct experiments employing binary and multi-class classification techniques for each dataset.

Binary classification

In this subsection, we present the evaluation results for binary classification scenarios using FL. In addition, we provide an evaluation of blockchain-based SSI.

1. Federated learning evaluation results. We employed 150,000 samples for both benign and attack instances in both datasets, ensuring a balanced dataset for a comprehensive and meaningful comparison. Remarkably, our model demonstrates impeccable performance, achieving perfect scores of 100% across all metrics for the Edge-IIoT dataset. In contrast, the results for the CICIoT2023 dataset remain highly promising, with an accuracy of 99.09%, indicating a low error rate in classifying both benign and malicious traffic. Furthermore, achieving a perfect precision of 100%, along with a recall of 98% and an F1-score of 99%, underscores the robust overall performance of the model, as shown in Fig. 8.

Figure 8 Evaluation of performance in binary classification.

The classification performance of our model is depicted through the confusion matrix presented in Fig. 9, providing a concise summary of the model’s accurate and erroneous predictions The primary goal is to minimize both false positive and false negative rates, ensuring precise classification outcomes. Our proposed model effectively achieves this objective, exhibiting false positive and negative rates of 0% in the Edge-IIoTset dataset. For the CICIoT2023 dataset, we observe a negligible false negative rate of 0.0017%, alongside false positive rates of 0. 96%, which confirms the accuracy and efficiency of the model in mitigating classification errors.

Figure 9 Confusion matrix in binary classification.

Furthermore, the receiver operating characteristic (ROC) curve and the area under the curve (AUC) depicted in Fig. 10 offer a visual representation of our model’s ability to distinguish between classes to highlight the model’s effectiveness, we achieve an AUC of 99.1% for CICIIoT2023 notably 100% for the EdgeIIoTset dataset.

Figure 10 ROC curve AUC in binary classification.

The training and validation loss curves for both the EdgeIIoTset and CICIoT2023 datasets demonstrate the promising performance of our federated learning model in binary classification. As illustrated in Fig. 11, the models exhibit rapid convergence within the initial epochs, followed by stable performance. The close alignment of training and validation loss curves, particularly in later epochs, indicates good generalization without significant overfitting. Both models achieve remarkably low final loss values (¡ 0.02) for training and validation sets, suggesting high predictive accuracy. The EdgeIIoTset model shows a slightly lower final loss, while the CICIoT2023 model displays smoother convergence between training and validation losses. These results collectively suggest that our approach effectively captures the underlying patterns in both datasets, promising strong performance on unseen data in real-world applications.

Figure 11 Train and validation loss of one local model in binary classification.

2. SSI-based DID evaluation results. As illustrated in Fig. 12 below both datasets exhibit similar startup durations (SD), with EdgeIIoTset demonstrating a marginally quicker performance by 0.01 s. Furthermore, the EdgeIIoTset dataset shows a shorter connect duration (CD) of 0.02 s compared to the CICIoT2023 dataset. Both datasets share the same publish duration (PD) of 9.15 s. However, in terms of issuing credential duration (ICD), the EdgeIIoTset dataset outperforms the CICIoT2023 dataset by 0.87 s. Regarding completed credential exchange duration (CCED), the EdgeIIoTset dataset exhibits a reduction of 8.9 s compared to the CICIoT2023 dataset. the CCED should normally be higher because this is the mutual authentication phase since the FS must authenticate all the nodes, and each node must authenticate the FS. Additionally, the average time per credential duration (ATCD) is shorter for the ‘EdgeIIoTset’ dataset in comparison to the ‘CICIoT2023’ dataset. Finally, the average time per transaction duration (ATTD) is marginally higher for the ‘EdgeIIoTset’ dataset when contrasted with the ‘CICIoT2023’ dataset.

Figure 12 Comparative performance analysis of blockchain-based SSI for binary classification.

Multiclass classification

In this subsection, we present the evaluation results for both the CICIoT2023 and EdgeIIoTset datasets in multiclass classification scenarios using FL. Additionally, we provide an evaluation of SSI-based DID.

1. Federated learning evaluation results. The evaluation of both the CICIoT2023 and Edge-IIoTset datasets reveals strong performance across diverse classes, as demonstrated in Table 4. Within the CICIoT2023 dataset, each class achieves good performance in most metrics. Notably, the DDoS, DoS, and Mirai attack classes in the CICIoT2023 dataset exhibit great classification capabilities, demonstrating perfect performance across all metrics with complete precision, recall, and F1-scores of 100%. Furthermore, the Spoofing, Benign, and Information Gathering classes show good precision with 87%, 83%, and 81%, respectively. However, their recall and F1 scores vary. The Benign class achieves a high recall of 85% and an F1-score of 84%. The Information Gathering class presents a relatively good recall of 84% and a corresponding F1-score of 83%. The Spoofing class achieves a moderate recall at 80% and an F1-score of 84%.

Table 4 Classification report.

Dataset	Class	Precision	Recall	F1-score	
CICIoT2023	Benign	83%	85%	84%	
	DDoS	100%	100%	100%	
	DoS	100%	100%	100%	
	Mirai	100%	100%	100%	
	Information gathering	81%	84%	83%	
	Spoofing	87%	80%	84%	
Edge-IIoTset	DoS/DDoS	91%	94%	92%	
	Information gathering	88%	88%	88%	
	Injection	77%	91%	84%	
	Malware attacks	97%	50%	74%	
	Man in the middle	100%	100%	100%	
	Normal	100%	100%	100%	

Transitioning to the Edge-IIoTset dataset Man in the Middle, and Normal classes also exhibit high performance, achieving perfect precision, recall, and F1-scores of 100%. followed by the DoS/DDoS class with high precision 91% and very high recall 99%, resulting in a strong F1-score of 92%. Furthermore, other classes show more variability. The Information Gathering class has an excellent precision, recall, and f1-score with 88%. The Injection class shows a precision of 77% and a high recall of 91%, leading to an F1-score of 84%. The Malware Attacks class, despite its high precision of 97%, suffers from a low recall of 50%, resulting in a lower F1-score of 74%.

The confusion matrix depicted in Fig. 13 provides valuable insights into the prediction frequency for each class compared to their actual occurrences. Within the CICIoT dataset, remarkable performance was observed for classes such as DoS, DDoS, and Mirai, with the model accurately predicting all samples, achieving a flawless prediction rate of 100%. Additionally, the classification of Reconnaissance exhibited high accuracy, with 84% of samples correctly classified followed by Benign traffic with 85%. However, in subsequent classes such as Spoofing, accuracy decreased, with only 80% respectively, of the samples accurately classified. Likewise, within the Edge-IIoTset dataset, classes like Man in the Middle, and Normal traffic demonstrated robust performance, as the model accurately predicts all samples, resulting in a 100% prediction rate. The classification of the DoS/DDoS, Injection, and Information gathering class followed suit, with 94%, 91%, and 88% respectively of samples correctly classified. Nevertheless, as we delve into subsequent classes such as Malware, declined, with only 50%, respectively, of samples accurately classified.

Figure 13 Confusion matrix in multiclass classification.

The ROC curves and AUC values for both datasets demonstrate excellent detection capabilities across various attack types as presented in Fig. 14. In the EdgeIIoTset dataset, most attacks show very good performance with AUC values above 0.90, with “Man in the middle” and “Normal” attacks achieving perfect detection (AUC 1.00). While “Malware” has the lowest performance (AUC 0.75), other attacks like “DoS/DDoS”, “Information gathering”, and “Injection” exhibit strong performance (AUC 0.96-0.93-.94). The second CICIoT2023 dataset similarly shows outstanding results, with DDoS, DoS, and Mirai attacks reaching perfect detection (AUC 1.00). “Benign” and “Recon” categories perform very well (AUC 0.90-0.91), and even the lowest performing “Spoofing” category maintains good detection ability (AUC 0.88). Notably, all attack types in both datasets significantly outperform the random guessing baseline, indicating robust and effective detection across the board.

Figure 14 ROC Curve AUC in multiclass classification.

Figure 15 illustrates the training and validation loss curves for both the EdgeIIoTset and CICIoT2023 datasets demonstrating the promising performance of our federated learning model in multi-class classification. For the EdgeIIoTset, we observe a rapid initial decrease in both training and validation loss, followed by a gradual convergence. The CICIoT2023 dataset shows a more gradual, consistent decrease in both losses across epochs. Importantly, neither dataset exhibits signs of overfitting, as the validation loss continues to decrease alongside the training loss, with only minimal divergence in later epochs. The EdgeIIoTset model achieves slightly lower final loss values (around 0.22) compared to the CICIoT2023 model (about 0.24), suggesting robust performance across different IoT datasets. The close alignment between training and validation losses, particularly in later epochs, indicates good generalization capabilities of our federated learning approach. These results suggest that our model effectively learns from local data without compromising privacy while maintaining strong predictive performance across diverse IoT classification tasks.

Figure 15 Train and validation loss of one local model in multiclass classification.

2. SSI-based DID evaluation results. As illustrated in Fig. 16 both datasets exhibit similar startup durations (SD), with EdgeIIoTset demonstrating a slight advantage of 0.08 s. The CICIoT2023 dataset shows a shorter connect duration (CD) by 0.01 s compared to EdgeIIoTset. Additionally, both datasets share the same publish duration (PD) of 9.15 s. However, the CICIoT2023 dataset boasts a shorter issuing credential duration (ICD) of 0.29 s compared to EdgeIIoTset. Regarding the completion of the credential exchange duration (CCED), the CICIoT2023 dataset surpasses EdgeIIoTset by 5 s. The CCED should normally be higher because this is the mutual authentication phase since the FS must authenticate all the nodes, and each node must authenticate the FS. Furthermore, the CICIoT2023 dataset demonstrates a shorter average time per credential duration (ATCD) by 0.36 s compared to EdgeIIoTset. Lastly, the CICIoT2023 dataset shows a shorter average time per transaction duration (ATTD) by 3 s compared to the EdgeIIoTset.

Figure 16 Comparative performance analysis of blockchain-based SSI for multiclass classification.

In Table 5 we provide a comparison between the performance of our work with other FL-based state-of-the-art IDS. The proposed SA-FLIDS demonstrates superior performance compared to existing state-of-the-art FL-based IDS approaches. It achieves the highest accuracy of 100% for binary classification on the EdgeIIoTSet dataset with a standard deviation σ = 0.00%, outperforming previous methods. For multiclass classification, SA-FLIDS attains 93.48% accuracy and σ = 0.12% on EdgeIIoTSet and 92% and σ = 0.47% on CICIoT2023, surpassing earlier works. The model’s consistently high performance across different datasets containing emerging new cyber attacks on IoT networks and classification tasks underscores its robustness and effectiveness in intrusion detection for IoT environments.

Table 5 Comparisons between SA-FLIDS and State-of-the-art works.

The bold values represent our method's performance.

FL-based IDS	Dataset	Classifier	Accuracy (%)	
Begum et al. (2024)	EdgeIIoTset binary	BiLSTM	96	
EdgeIIoTset multiclass	BiLSTM	83	
EdgeIIoTset binary	CNN	85.31	
EdgeIIoTset multiclass	CNN	97	
Baucas, Spachos & Plataniotis (2023)	Human activity recognition	CNN	91.75	
Chatterjee & Hanawal (2021)	NSL-KDD	MLP	88	
Schneble & Thamilarasu (2019)	PhysioNet	ANN	99	
Ashraf et al. (2022)	Ba-IoT binary	ANN	99	
Our	EdgeIIoTSet binary	LSTM	100, ( σ = 0.00)	
	EdgeIIoTSet multiclass	93.48, ( σ = 0.12)	
	CICIoT2023 binary	99.12, ( σ = 0.02)	
	CICIoT2023 multiclass	92, ( σ = 0.47)	

Limitation of the experimental design

Scalability considerations: Our experiments involved a relatively small number of fog nodes ( clients K = 10). While this setup demonstrated the effectiveness of our approach, it may not fully represent the scalability challenges in larger, more complex IoMT networks. Future work should explore the performance and efficiency of SA-FLIDS in larger-scale deployments.

IID data assumption: Our current implementation assumes Independent and Identically Distributed (IID) data across clients. This assumption may not hold in all real-world scenarios, potentially impacting the model’s performance in non-IID settings. Further investigation into non-IID data distributions is necessary.

Communication overhead and system performance

Our proposed architecture employs several strategies to minimize communication overhead and optimize system performance. The integration of IoMT devices and DIDs in the fog computing environment has been carefully designed to reduce network load: Data exchange optimization: IoMT devices share data only during the initialization phase, with subsequent communications limited to updates. This approach significantly reduces the volume of data transferred across the network.

Efficient authentication: The system utilizes a session-based authentication mechanism. A token is generated once per session, eliminating the need for repeated DID authentications and thereby reducing associated overheads.

Two-stage communication process: The system operates in two distinct stages: (a) Initial authentication using VC and DID communication.

(b) Subsequent update exchanges using the authenticated token.

This separation ensures that DID communication and network protocol operations do not run concurrently, further optimizing resource usage.

Blockchain utilization: Blockchain’s role is to provide authentication support only, handling DID and VC registration, verification, and revocation. It does not store FL data. Consequently, its workload is limited to operations during the initialization phase (allocation of DIDs), the authentication phase (creation and verification of VCs), and the node revocation phase (revocation of VCs). Furthermore, no mining operations, which are known to consume a lot of energy and computing power, are carried out. The only situation requiring additional work is when a new node is added to the network, which significantly reduces its workload and associated overheads.

Ensuring privacy and FL model implementation: What’s special about DIDs is that they operate on the same principle as SSL/TLS certificates. Obtaining a DID implies having a pair of public and private keys stored in a wallet, which is used to guarantee the confidentiality of exchanges.

Figure 17 illustrates the total communication overhead for binary and multiclass classification tasks using the CICIoT2023 and EdgeIIoTset datasets. The chart demonstrates that multiclass classification requires significantly higher communication overhead compared to binary classification for both datasets. Interestingly, while the overhead for binary classification is identical (14.64 MB) for both datasets, there is a slight difference in multiclass classification, with CICIoT2023 requiring marginally more overhead (66.05 MB) than EdgeIIoTset (64.49 MB). This comparison provides insights into the computational demands of different classification tasks and datasets in federated learning-based intrusion detection systems.

Figure 17 Comparison of communication overhead.

Security and privacy analyses

Our research focuses on detecting potential attacks on smart healthcare systems enabled by IoMT networks. The aim is to mitigate the risks associated with unauthorized access to sensitive patient data through malicious IoMT devices. Thus, in this section, we thoroughly investigate the SA-FLIDS model’s privacy and security features. It is important to note that without appropriate security measures, people may be reluctant to participate in healthcare applications, which ultimately hinders the success of these technological advancements. Our SA-FLIDS framework addresses these concerns by providing robust security measures, instilling confidence in users, and promoting widespread adoption and sustainability of healthcare technologies. Additionally, the analysis process is rooted in a theoretical exploration of SA-FLIDS’s resilience against potential attacks outlined in the adversary model (“Detectin Process FL in SA-FLIDS system”).

Table 6 provides a detailed comparative analysis between the existing systems and our proposed model. In this analysis, we examine the security commitments of the SA-FLIDS model compared to other FL-based IDS. We also compare the SA-FLIDS system with the broader context of security considerations in FL. Notably, the SA-FLIDS model advances it further by incorporating additional layers of security, such as user authentication and communication channels security during the FL process, by using a reliable aggregation technique. Furthermore, our comprehensive approach enhances the level of sensitive data protection and stands out as the only scheme incorporating SSI for user authentication in the FL process. This feature strengthens the system’s security posture by ensuring authorized access and preventing unauthorized participation.

Table 6 Comprehensive comparison between existing works and our model.

Models	FL	IDS	Environment	Security techniques of FL	
				Against adversarial	Secure communication	Blockchain	SSI authentication	
Schneble & Thamilarasu (2019)	✓	✓	Medical CPS	X	X	X	X	
Chatterjee & Hanawal (2021)	✓	✓	IoT network	X	X	X	X	
Man et al. (2021)	✓	✓	IoT network	X	X	X	X	
Rey et al. (2022)	✓	X	IoT network	✓	X	✓	X	
Ruzafa-Alcázar et al. (2021)	✓	✓	Industrial IoT network	✓	✓	X	X	
Zhao et al. (2019)	✓	X	General purpose	X	X	X	X	
Friha et al. (2022)	✓	✓	Agriculture IoT	X	✓	✓	X	
Ashraf et al. (2022)	✓	✓	Healthcare IoT network	✓	X	✓	X	
Preuveneers et al. (2018)	✓	✓	Industrial IoT network	✓	X	✓	X	
Lakhan et al. (2022)	✓	X	Healthcare IoT network	X	X	✓	X	
OUR MODEL	✓	✓	Healthcare IoT network	✓	✓	✓	✓	

Data privacy and security analysis

SA-FLIDS leverages a fog-based NIDS powered by FL for data privacy-preserving. This allows the NIDS to effectively classify and detect malicious network traffic in real-time, protecting patient privacy and health data confidentiality. On the other hand, Also, the SA-FLIDS system deploys its security shield close to the source, which leads to a seamless granting of access to normal traffic while automatically blocking malicious intrusions, ensuring a secure and trustworthy healthcare environment.

Moreover, our system ensures FL protection through the implementation of blockchain-based SSI technologies. These technologies play a crucial role in securing the system against authorization and privacy concerns using the DID and VC techniques. By leveraging the immutability and integrity features of blockchain, it becomes practically impossible for any entity to manipulate, replace, or falsify user identities stored on the blockchain.

Analysis of FL attacks

Table 7 provides adversarial attack scenarios targeting the FL process and how our models resist those attacks.

Table 7 Summary of attack scenarios and countermeasures.

Adversarial attacks scenario on FL	Description	Countermeasures	
Sybil attacks	Adversary creates multiple fake identities to disrupt the FL process.	Utilizes blockchain-based DIDs and VCs for unique identification and authentication of nodes, preventing fake identities.	
Data poisoning attacks	Adversary injects malicious data to degrade global model performance.	Employs trimmed mean aggregation to minimize the influence of outliers and malicious data.	
Eavesdropping and data tampering attacks	Unauthorized interception or modification of communication data.	Ensures secure communication using gRPC framework with TLS for end-to-end encryption.	
Unauthorized access and authentication attacks	Adversaries gain unauthorized access by exploiting weak authentication.	Incorporates blockchain-based SSI technologies to ensure that only authenticated devices can participate in the FL process.	

Analysis of cyber-attacks in healthcare systems

After applying the countermeasures, we achieve a secure and authenticated FL (SA-FL) system. This SA-FL system is then integrated into an IDS to identify and mitigate cyber-attacks on IoMT network traffic, specifically for healthcare applications. Resisting a range of attacks such as DoS, DDoS, Information Gathering, Web-Based Vulnerabilities, Communication Spoofing, Brute-Force, and Mirai IoT threats. The model ensures data security and integrity by preventing unauthorized access, maintaining fog server access, and safeguarding patient lives. By implementing robust security measures, SA-FLIDS promotes the sustainability of the smart healthcare system by fostering user adoption and confidence in healthcare technologies.

Conclusion and future work

This article introduces SA-FLIDS, a Secure and Authenticated Federated Learning-based Network Intrusion Detection System designed for Fog-IoT-enabled smart healthcare systems. Our research aims to detect and counter cyber attacks on IoMT by harnessing fog computing capabilities. Additionally, we aim to preserve data privacy and reduce communication overhead, while addressing vulnerabilities like poisoning and Sybil attacks inherent in decentralized FL paradigms. We achieve this by employing a blockchain-based SSI model for client authentication and using trimmed mean aggregation in FL. In addition, secure communication transfer is ensured through TLS and gRPC protocols. Performance evaluation demonstrates that SA-FLIDS not only detects attacks on the Internet of Medical Things (IoMT) but also meets criteria for privacy preservation, scalability, and sustainability. Furthermore, Our SA-FLIDS framework achieves high accuracy with negligible false positives and false negatives, particularly in binary classification scenarios. Our future endeavors will focus on evaluating the performance of our proposed model across various domains of IoT applications. Additionally, we aim to explore the application of FL with non-distributed IID data distributions.

Additional Information and Declarations

Competing Interests

Author Contributions

Data Availability

Leandros Maglaras is an Academic Editor for PeerJ.

Radjaa Bensaid conceived and designed the experiments, performed the experiments, analyzed the data, performed the computation work, prepared figures and/or tables, authored or reviewed drafts of the article, and approved the final draft.

Nabila Labraoui conceived and designed the experiments, authored or reviewed drafts of the article, and approved the final draft.

Ado Adamou Abba Ari performed the experiments, authored or reviewed drafts of the article, and approved the final draft.

Hafida Saidi analyzed the data, performed the computation work, prepared figures and/or tables, authored or reviewed drafts of the article, and approved the final draft.

Joel Herve Mboussam Emati analyzed the data, authored or reviewed drafts of the article, and approved the final draft.

Leandros Maglaras conceived and designed the experiments, authored or reviewed drafts of the article, and approved the final draft.

The following information was supplied regarding data availability:

The source code (complete implementation of the SA-FLIDS framework, including the Federated Learning process, the blockchain-based Self-Sovereign Identity (SSI) component, and the necessary scripts for, model training, and evaluation) for the SA-FLIDS framework is available at Zenodo: teamflssi. (2024). teamflssi/Secure-and-Authenticated-Federated-Learning-based-intelligent-NIDS-for-smart-healthcare: v1.4 (v1.3). Zenodo. https://doi.org/10.5281/zenodo.11316425.

The raw data used for experimental results is available at Zenodo: STIC Laboratory. (2024). preprocessed datasets. Zenodo. https://doi.org/10.5281/zenodo.11315294.

The third-party datasets are available at:

- CICIoT2023 dataset: (Neto, Dadkhah, and Ferreira 2023), https://www.unb.ca/cic/datasets/iotdataset-2023.html.

- Edge-IIoTset dataset: (Ferrag et al., 2022), https://www.kaggle.com/datasets/mohamedamineferrag/edgeiiotset-cyber-security-dataset-of-iot-iiot.

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
