# Peer review of "SA-FLIDS: secure and authenticated federated learning-based intelligent network intrusion detection system for smart healthcare"

_PeerJ Computer Science, doi:10.7717/peerj-cs.2414_

## Round 0.1 · original submission · Major Revisions

The review process is now complete. While finding your paper interesting and worthy of publication, the referees and I feel that more work could be done before the paper is published. My decision is therefore to provisionally accept your paper subject to major revisions.

Reviewer 1 ·

Basic reporting

The paper presents a novel framework called Secure and Authenticated Federated Learning-based Intelligent Network Intrusion Detection System (SA-FLIDS) tailored for smart healthcare environments. It combines federated learning with blockchain technology to enhance data privacy, integrity, and security, mitigating risks associated with centralized data storage. The proposed system demonstrates strong resilience against adversarial attacks and is evaluated using real IoT traffic datasets, showing significant improvements in accuracy and security over existing methods. The study highlights the framework's potential for real-world applications in securing Internet of Medical Things (IoMT) networks.

There are instances where complex sentences could be simplified to improve readability. For example, breaking down long sentences into shorter, more digestible parts can help in conveying complex information more clearly.

Consider revising sentences such as "The integration of blockchain technology with federated learning provides a decentralized and secure environment for intelligent NIDS, thus enhancing data privacy and integrity while mitigating the risks associated with centralized data storage.

The writing style is formal and adheres to academic standards, which is suitable for a scholarly article.

Experimental design

The experimental setup could benefit from a more detailed description of the hardware and software environments used.

It would be helpful to include a discussion on the limitations of the experimental design and potential impacts on the results.

Validity of the findings

While the manuscript mentions the use of statistical tests, there is a lack of detailed explanation regarding the specific tests employed and the rationale behind their selection. Providing more information on these tests would help readers better understand the statistical rigor of the study.

The manuscript does not adequately discuss potential biases in the datasets used and how these biases might impact the findings. A more detailed discussion on this aspect would enhance the credibility of the results.

It would be beneficial to include information on how representative the datasets are of typical IoMT traffic and whether there are any limitations in their generalizability.

Reviewer 2 ·

Basic reporting

- It would be better for the quality of the paper to give motivation in the introduction section with a separate subsection.
- Providing a detailed flow diagram of the architecture would improve the quality of the paper.
- More literature should be included on federated learning-based blockchain systems in IoMT.
- Figure 2 is important to convince the reader of the architecture concept. It would be better to specify where the data comes from and how it is encrypted.

Experimental design

- How does the communication cost of IoMT and DID used in the fog node affect the overhead on the network?
- What kind of overhead does the communication protocol used in the network impose on the DID communication overhead on the system?
- Additionally, the authentication of the DID record on the blockchain affects the blockchain overhead.
The system's overhead affects performance. To convince the reader, it would be better to explain it in detail.
- It would be better for the quality of the paper to compare the communication overhead of the protocols used in the network with other architectures.

Validity of the findings

- How to ensure privacy until the IoMT raw data is transmitted to the fog node and fog server, and how the FL model is realized in the fog server. A detailed explanation would improve the quality of the paper.
- The security analysis appears weak. More detail with attack scenarios would be better regarding the quality of the paper.

Reviewer 3 ·

Basic reporting

Authors have proposed a new framework for intrusion detection in this paper. However, there are major concerns as follows:

- The literature review presented by the authors is not up-to-date. They need to include studies from the years 2022-2023.
- The authors have listed their contributions point by point in the paper. However, it is not clear what the exact contribution of the paper is. For example, is "Using a blockchain-based SSI model in FL" the first study to do so? Additionally, the sentence "Exploring the application of a Federated Learning-based Network Intrusion Detection System (NIDS) for analyzing network traffic, identifying potential attacks, and ensuring privacy in IoMT-based smart healthcare applications on distributed fog networks" is not a contribution.

Experimental design

- Providing the validation and training loss values is important for understanding the training process of the proposed deep learning model.

Validity of the findings

- It is recommended to use the AUC parameter as an evaluation metric. Additionally, providing the average accuracy metric and standard deviation values obtained during the 5-fold validation process is essential for an accurate assessment of the model's performance.
- The authors have not compared their classification performance with any similar studies. This is crucial for understanding the contribution of the work to the literature. Including a comparison with a recent study will better demonstrate the model's contribution to the literature.

---

## Round 0.2 · accepted · Accept

We are happy to inform you that your manuscript has been accepted since the reviewers' comments have been addressed.

Reviewer 1 ·

Basic reporting

Authors updated the paper and no further update needed from my side.

Experimental design

As above

Validity of the findings

As above

Reviewer 2 ·

Basic reporting

The authors have addressed my comments and concerns in their revised manuscript.

Experimental design

The authors have addressed my comments and concerns in their revised manuscript.

Validity of the findings

The authors have addressed my comments and concerns in their revised manuscript.

Reviewer 3 ·

Basic reporting

The authors completed all revision processes successfully.

Experimental design

The authors completed all revision processes successfully.

Validity of the findings

The authors completed all revision processes successfully.